# A Deep Learning Network to Retrieve Ocean Hydrographic Profiles from Combined Satellite and In Situ Measurements

**Bruno Buongiorno Nardelli** 

Consiglio Nazionale delle Ricerche, Istituto di Scienze Marine (CNR-ISMAR), 80133 Naples, Italy;
bruno.buongiornonardelli@cnr.it

**Abstract:** An efficient combination of remotely-sensed data and in situ measurements is needed to obtain accurate 3D ocean state estimates, representing a fundamental step to describe ocean dynamics and its role in the Earth climate system and marine ecosystems. Observations can either be assimilated in ocean general circulation models or used to feed data-driven reconstructions and diagnostic models. Here we describe an innovative deep learning algorithm that projects sea surface satellite data at depth after training with sparse co-located in situ vertical profiles. The technique is based on a stacked Long Short-Term Memory neural network, coupled to a Monte-Carlo dropout approach, and is applied here to the measurements collected between 2010 and 2018 over the North Atlantic Ocean. The model provides hydrographic vertical profiles and associated uncertainties from corresponding remotely sensed surface estimates, outperforming similar reconstructions from simpler statistical algorithms and feed-forward networks.

**Keywords:** artificial intelligence; machine learning; deep learning; neural networks; Earth observations; ocean dynamics; sea surface temperature; sea surface salinity; altimetry; hydrography

## 1. Introduction

Remote monitoring of ocean processes represents a relevant and challenging goal, as ocean dynamics comprise several processes (inter)acting over a wide range of spatial and temporal scales, which may influence the Earth climate and drive significant marine ecosystem changes. Notably, several crucial processes contributing to the transport of momentum, energy, chemicals and marine organisms cannot be fully understood unless repeated views of the 3D ocean state and surface forcings are available. This is particularly relevant for processes in the ocean mesoscale to sub-mesoscale range, given their intrinsic 3D nature [1–3]. In turn, the dynamical response and feedbacks of these processes to natural and anthropogenic pressures also remains largely uncertain. However, given both the theoretical and practical limitations of available technologies, observations can only provide partial views of the ocean state, especially if analysed separately. Ingenious approaches are thus needed to project, at depth, remotely sensed data acquired from instruments looking at the sea surface from space, taking advantage of the sparse in situ measurements collected throughout the water column.

Scientists have followed two main complementary approaches to project surface data at depth and provide a description of the 3D ocean state: the assimilation of observations in numerical models and the combination of purely data-driven reconstructions and diagnostic models. Both strategies are affected by strengths and weaknesses, though.

The data assimilation in prognostic models can guarantee the ocean state to evolve in a consistent way with the physics represented by the model [4–6]. Models, however, are affected by uncertainties in initialization and forcings, and need parameterizations of sub-grid scale processes, which may lead

to inaccurate representations of the physics, especially when aiming to reconstruct long timeseries for decadal/climatological studies (due to grid size limitations and subsequent need to parameterize also mesoscale processes—e.g., [7]). In general, models' abilities to reproduce non-assimilated observations is further hindered by the difficulty to properly account for model and observation representativeness and errors.

Data-driven approaches are based on a synergic use of different satellite, in-situ measurements and diagnostic models. They can reduce the differences between reconstructed and independent observations (for 2D examples see [8–10]), but usually allow only a much simpler description of the dynamics with respect to general circulation models (sometimes limited to zero or first order balances, such as geostrophy and quasi-geostrophy, or simple Ekman models). Data-driven 3D reconstruction techniques that found a systematic application are based on purely empirical and/or statistical regressions/analyses [11–21], eventually coupled to dynamical diagnostic tools (for full 3D examples, see [22,23]). Methodologies derived within the surface quasi-geostrophy framework make much stronger assumptions on the ocean vertical stratification, though providing interesting theoretical perspectives [24–29]. More recently, mixed approaches have also been explored [30].

All data driven approaches share the objective to project surface information at depth, starting from synoptic satellite observations and some prior knowledge of the hydrography. Despite the recent advancements in machine learning algorithm implementation and a growing interest in the possibilities opened by artificial intelligence for data science, only few attempts have been carried out until now to address this specific objective with artificial neural networks (e.g., in [31–37]), either based on generalized regression neural networks, self-organizing maps, support vector machines, decision trees or feed-forward neural networks. Models based on neural networks have also been proposed to "augment" observed vertical profiles with variables that have not been directly measured (e.g., in [38–40]).

In this paper, a stacked Long Short-Term Memory network (LSTM, [41]) is coupled to a Monte Carlo dropout approach and used to project satellite surface data at depth after training with sparse co-located in situ vertical profiles. LSTM is a deep learning algorithm particularly suited to exploit sequential information as those present in hydrographic profiles. Dropout provides both a regularization strategy, when applied during training, and a "Bayesian" inference approximation if applied during both training and testing [42]. As such, the technique proposed here is able to provide both vertical hydrographic profiles and uncertainties on the predicted values.

This work was carried out within the European Space Agency World Ocean Circulation project (ESA-WOC), as a preparatory step for the development of a daily 3D reconstruction of the dynamics in the North Atlantic (down to 1500 m depth) at 1/10° spatial resolution, covering the period between 2010 and 2018. As such, the network was trained and tested, taking as target (output) the measurements collected by Argo profilers and CTD casts within a wide portion of the North Atlantic over that period. Co-located satellite-derived sea surface temperature, sea surface salinity and absolute dynamic topography values (extracted from operational and experimental products) were used as input data.

The performance of the proposed LSTM network has been assessed by keeping part of the in situ profiles as independent reference observations during test. Root mean squared errors were estimated from LSTM profiles, from climatological data and multivariate Empirical Orthogonal Function reconstructions (mEOF-r, as in [16]), as well as from the output of simpler feed-forward networks.

## 2. Materials and Methods

### 2.1. Data: Surface Measurements

The SST used in the present study was the level 4 (L4—i.e., interpolated) multi-year reprocessed Operational Sea Surface Temperature and Sea Ice Analysis (OSTIA) developed by the U.K. Met Office and distributed (upon free registration) through the Copernicus Marine Environment Monitoring Service (CMEMS, http://marine.copernicus.eu/services-portfolio/access-to-products/,

product_id=SST_GLO_SST_L4_REP_OBSERVATIONS_010_011). OSTIA combines the reprocessed ESA SST CCI, C3S, EUMETSAT, REMSS and OSPO satellite data, and in situ data from HadIOD and provides daily maps of foundation SST (i.e., not affected by the diurnal cycle). The analysis runs an optimal interpolation (OI) algorithm on a 1/20° regular grid [43]. OSTIA was sub-sampled here to 1/10° resolution, and resulting grid was also used for the pre-processing of the other surface datasets (see the SST example in Figure 1a).

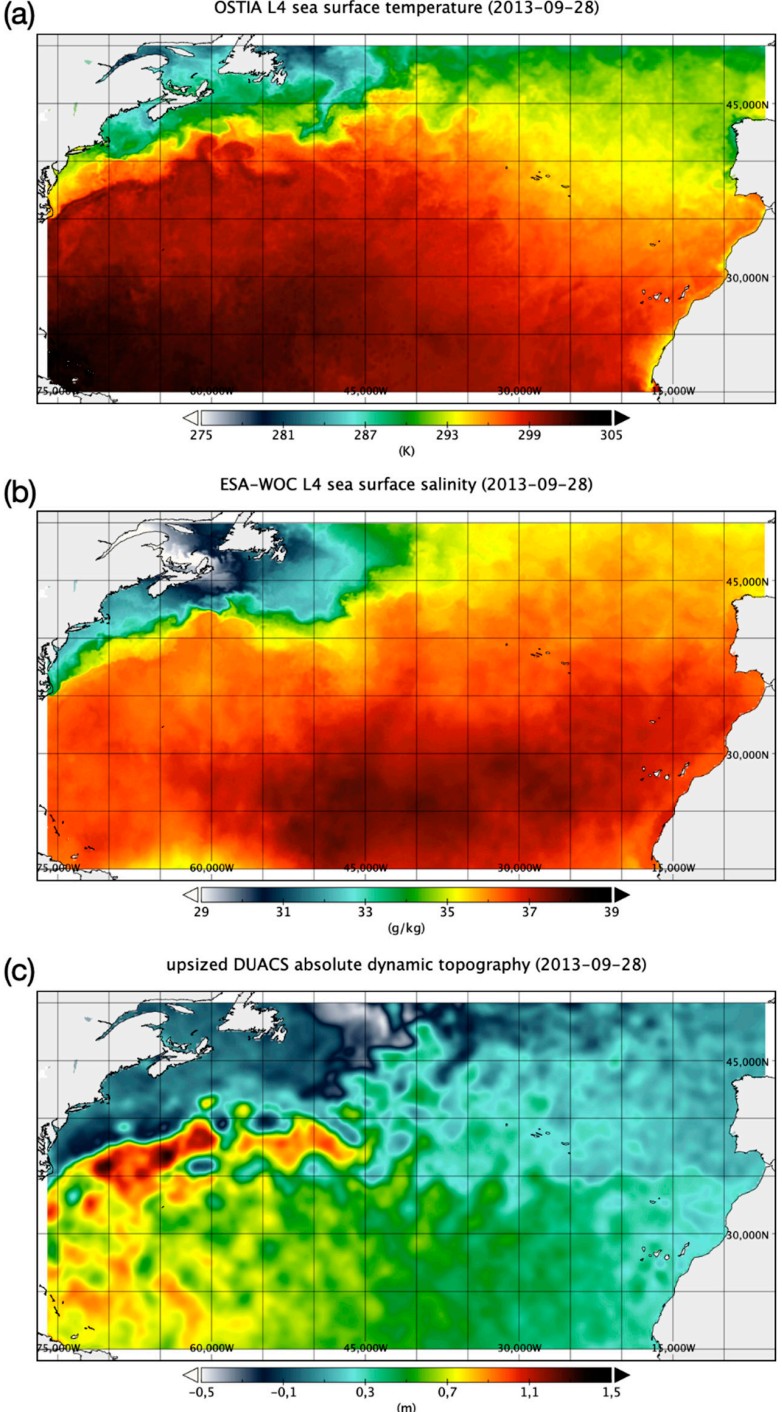

**Figure 1.** Examples of the surface daily data taken as input to the reconstruction techniques: OSTIA L4 reprocessed SST (**a**), SSS L4 developed within ESA-WOC project (**b**), adjusted ADT L4 derived from DUACS data (**c**).

The SSS data were developed within ESA-WOC project [44]. They were obtained by adapting to the 1/10° North Atlantic grid the multidimensional optimal interpolation algorithm used to retrieve CMEMS global dataset (http://marine.copernicus.eu/services-portfolio/access-to-products/, product_id: MULTIOBS_GLO_PHY_REP_015_002, dataset_id: dataset-sss-ssd-rep-weekly). This algorithm interpolates SMOS observations and in situ SSS observations considering a space–time-thermal decorrelation function, estimated by including information from high-pass filtered daily SST data [45,46]. This multivariate approach effectively increases the SSS resolution by using satellite SST differences to constrain the surface patterns (see Figure 1b). Here, we ingested the SMOS L3OS 2Q debiased daily salinity disseminated by the Centre Aval de Traitement des Données SMOS (CATDS, 2017), OSTIA SST data and CORA5.2 surface values (see Section 2.2) as input data, and used the CMEMS weekly SSS dataset to build our background field (linearly interpolating it in time between the 1/10° grid through a cubic spline). All other interpolation parameters were set as in [47].

The Absolute Dynamic Topography (ADT) data considered here were based on the altimeter Sea Level Anomaly (SLA) product provided by SSALTO/Data Unification and Altimeter Combination System (DUACS). They are obtained by adding a Mean Dynamic Topography [48] to the SLA field, and are distributed by CMEMS as reprocessed data (http://marine.copernicus.eu/services-portfolio/access-to-products/, product_id: SEALEVEL_GLO_PHY_L4_REP_OBSERVATIONS_008_047). ADT has been upsized here to the ESA-WOC 1/10° × 1/10° grid through a cubic spline (see example in Figure 1c). ADT data were pre-processed to make them consistent with in situ steric heights. The adjustment was carried out as in [16], namely by regressing steric heights and co-located ADT data in the neighbourhood of each grid point, considering matchups within a temporal window of ±10 days.

### 2.2. Data: In Situ Vertical Profiles

The vertical hydrographic profiles were taken from the quality controlled Argo and CTD profiles produced by CMEMS CORA 5.2 (http://marine.copernicus.eu/services-portfolio/access-to-products/, product_id: INSITU_GLO_TS_REP_OBSERVATIONS_013_001_b) [49]. The data considered here were to the 2010–2018 period, and were interpolated through a spline on a regularly spaced vertical grid (with 10 m intervals). Steric heights were computed taking 1500 m as reference level.

### 2.3. Data: Climatology

Temperature and salinity monthly climatological fields computed by the World Ocean Atlas 2013 were used to convert all daily observations to anomaly fields (see Section 3). These climatologies are estimated on a 1/4° × 1/4° grid by applying an objective analysis algorithm [50,51]. The values in the first 1500 m, provided on 125 levels, were interpolated through a spline on a regularly spaced vertical grid (with 10 m intervals), and upsized to the 1/10° ESA-WOC grid through a cubic spline.

### 2.4. Methods: Multivariate Empirical Orthogonal Function Reconstruction (mEOF-r)

The multivariate Empirical Orthogonal Function reconstruction (mEOF-r) was taken as reference for the retrieval of the 3D hydrographic fields. This methodology was applied in many previous studies [16,18,52–54], and it is thus only briefly recalled hereafter. It starts by building a state vector by concatenating (normalized and interpolated on a regular vertical grid) temperature, salinity and steric heights anomaly profiles, and decomposing its variability in EOF modes (thus called multivariate EOF—for an example of multivariate EOF modes in the North Atlantic, see Figure 3 in [18]). Anomalies are defined with respect to monthly WOA13 data (linearly interpolated in time between the central day of each month, and through a cubic spline horizontally). The EOFs were computed from available in situ observations, and the decomposition was truncated to a maximum of three modes. The three elements in the state vector reconstructed from the truncated EOF that correspond to the surface are equated to the anomalies of SST, SSS and adjusted ADT. In this way, a linear system is obtained, the unknowns being the three EOF amplitudes. Once solved through a

trivial matrix inversion, full profiles associated with each mode can be estimated and finally summed up to get the synthetic vertical reconstruction.

In order to account for local differences in the dynamics, the configuration proposed in [16] has also been adopted here, as detailed as follows. The North Atlantic domain is divided into subdomains with a maximum extension of 30° both in latitude and in longitude. Multivariate EOFs are estimated considering only the in situ profiles collected within ±20 days with respect to the reconstruction day. To remove eventual discontinuities in the reconstruction, all neighbouring subdomains are overlapped by one half of their latitudinal and longitudinal extensions. In the grid points where multiple reconstructions are available, these are averaged out by bilinearly weighting them with the inverse of the distance to each subdomain centre. In some cases, two modes (or even one mode) may be sufficient to retrieve most of the variability and the reconstruction error may increase if more modes are added (more than 95% of the variance is generally explained by the selected modes). Consequently, the optimal number of modes for the 3D reconstruction is chosen by evaluating the mean hindcast error within each subdomain, so as to minimize the root mean square difference between the input profiles and the synthetic profiles reconstructed from corresponding in situ surface measurements.

## 2.5. Methods: Feed-Forward Neural Networks

Feed-forward networks represent the simplest type of artificial neural networks and consist of one input layer (the input vector) and one output layer (the output vector) connected through a variable number of hidden layers (if that number is >1 we speak about a "deep network"). Each of the layers is made up by a variable number of units: the elements of the vectors in the case of the input/output layers, and the artificial "neurons" (or computing nodes) in the hidden layers. Each of the units in one layer is connected to all units in the following layer through weights that are estimated during the network training, and each computing node processes the sum of its weighted input by passing it through an activation function, which provides the neuron's output. FFNN networks are designed to model complex flows of information from the input to the output and are common candidates to solve non-linear regression problems. The definition of a proper model for each specific problem, however, requires optimizing the choice of several "hyper-parameters", starting from the number of hidden layers, the number of units within each hidden layer, to the activation function to apply within each hidden layer. For a given architecture, network training also implies a number of additional choices. In fact, training is performed by minimising some model loss functions, while iteratively feeding the network with several input–output samples. Various algorithms exist to this aim, and the same network trained in a different way on the same data may indeed lead to different results ("local" optima). Different results can be obtained depending also on the number of iterations (epochs) considered.

Large feed-forward networks (in terms of number of layers/units) are prone to over-fitting, as distinct sets of neighbouring neurons might adjust to reproduce individual samples, leading to complex co-adaptations which would not allow us to generalize the network to unseen data. Again, different strategies can be followed to avoid co-adaptation: the dropout approach followed here is described in Section 2.7.

In our tests, two different input/output vectors have been initially considered. In the first case, somehow imitating simple multilinear regression approaches, the input vector was made up of the target depth for the retrieval, the anomalies of SST, SSS and adjusted ADT (same as in mEOF-r), the latitude, and longitude, and the day of the year projected on a circle (as in [34], namely $day_1 = \cos\left(\frac{2\pi}{365} day\ of\ the\ year\right) + 1$, $day_2 = \sin\left(\frac{2\pi}{365} day\ of\ the\ year\right) + 1$), while the output included the co-located values of temperature, salinity and steric heights anomalies at the target depth. In the second configuration, the depth was dropped from the input data and the concatenated temperature, salinity and steric heights anomaly profiles were taken as output (same as mEOF-r state vector). All vectors was preliminary normalized through min–max algorithm (namely rescaling the range of features to scale within the [0, 1] range).

Several preliminary hyper-parameter tuning tests were carried out, considering one to three hidden layers, and a variable number of units within each, ranging between 5 and 50, with a 5-unit increase step. The sigmoid performed better than the hyperbolic tangent as an activation function. "*Adam*" was selected as the network optimizer, taking the mean squared error as loss metrics. In total, 15% of input samples were randomly kept as a holdout validation dataset at each iteration. The number of optimal training epochs was found by monitoring model performance (early stopping).

The performance of the first set of models (those retrieving values at individual depths) never improved with respect to the climatology, while a visible improvement was found in the second configuration, especially when the choosing two hidden layers. The successive tests were thus restricted to this latter configuration, significantly increasing the number of hidden units (tests were run with up to 5000 units per layer). The final FFNN architecture considered (which further improved the reconstruction accuracy) included 1000 units in each of the two hidden layers, (hereafter, FFNN (1000–1000)). Above that, the number of hidden units and performance substantially stabilized.

## 2.6. Methods: Long Short-Term Memory Networks

Recurrent neural networks (RNN) can be described as sequences of sub-networks (also called "cells") designed to include information from the previous cell in a sequence as input to the successive one. This makes them particularly fit to model ordered sequences of data. Simple recurrent networks, however, are not able to efficiently process information from cells that lie too far along the sequence, due to vanishing/exploding values in the gradient-descent based optimizations.

Long Short-Time Memory (LSTM) network is a particular type of RNN, that is specifically designed to avoid vanishing/exploding gradients and preserve the relevant information flow throughout the network [41]. Within LSTM cells, the external input vector ($x_i$) is concatenated to the previous cell hidden state ($h_{i-1}$) and then passed through different "gates", each one aimed at carrying out a specific task to update both the hidden state itself ($h_i$) and a cell state ($C_i$), that is directly transmitted to the next cell and basically acts as a network "memory". The LSTM cell specifically includes a forget gate, an input gate, and an output gate, as depicted in Figure 2a: whose equations thus read:

$$f_i = \sigma\left(W_f[h_{i-1}, x_i] + b_f\right) \tag{1}$$

$$I_i = \sigma(W_I[h_{i-1}, x_i] + b_I) \tag{2}$$

$$\widetilde{C}_i = tanh(W_C[h_{i-1}, x_i] + b_C) \tag{3}$$

$$O_i = \sigma(W_O[h_{i-1}, x_i] + b_O) \tag{4}$$

$$C_i = f_i * C_{i-1} + I_i * \widetilde{C}_i \tag{5}$$

$$h_i = O_i * tanh(C_i) \tag{6}$$

where $\sigma$ and *tanh* represent the sigmoid and hyperbolic tangent activation functions, respectively, and $W_x$ and $b_x$ represent model weights and biases.

LSTM networks can include a single layer of LSTM cells or multiple LSTM layers stacked one on top of the other, potentially leading to quite deep architectures. The number of cells in each layer matches the length of the sequence by definition, but the number of hidden units still needs to be configured.

Here, the sequential information to exploit is provided by a multivariate output state vector comprising temperature, salinity and steric height anomaly profiles. In practice, each cell in the sequence considers in input the same values (i.e., the anomalies of SST, SSS and adjusted ADT, plus latitude, longitude and cyclic day as in previous section), but takes the output values at increasing depths (with depth "acting" as time in more standard applications of LSTM). As for FFNN models, all vectors are scaled within the 0–1 range before feeding the network.

The number of hidden units considered ranged between 5 and 50, with a 5-unit increase step, and three different network architectures were tested: a simple LSTM and two stacked LSTMs (with 2 and 3 layers each). The optimization algorithm and related parameters were exactly the same as those used for the FFNN reconstruction training, as well as the dropout strategy applied to avoid overfitting and obtain reconstructed profile uncertainties (see next Section).

The best performance was obtained with a 2-layer stacked network, including 35 hidden units in each LSTM layer (hereafter, LSTM (35–35)), as depicted in Figure 2b.

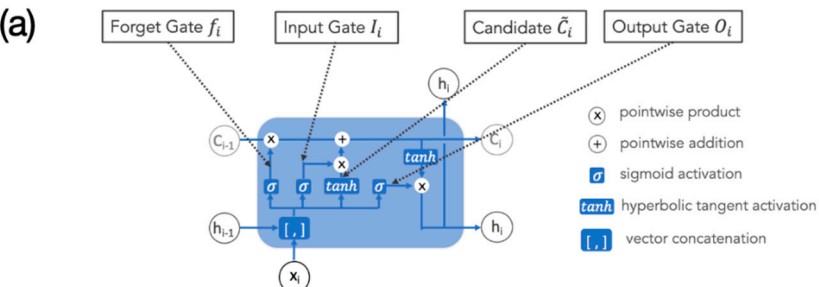

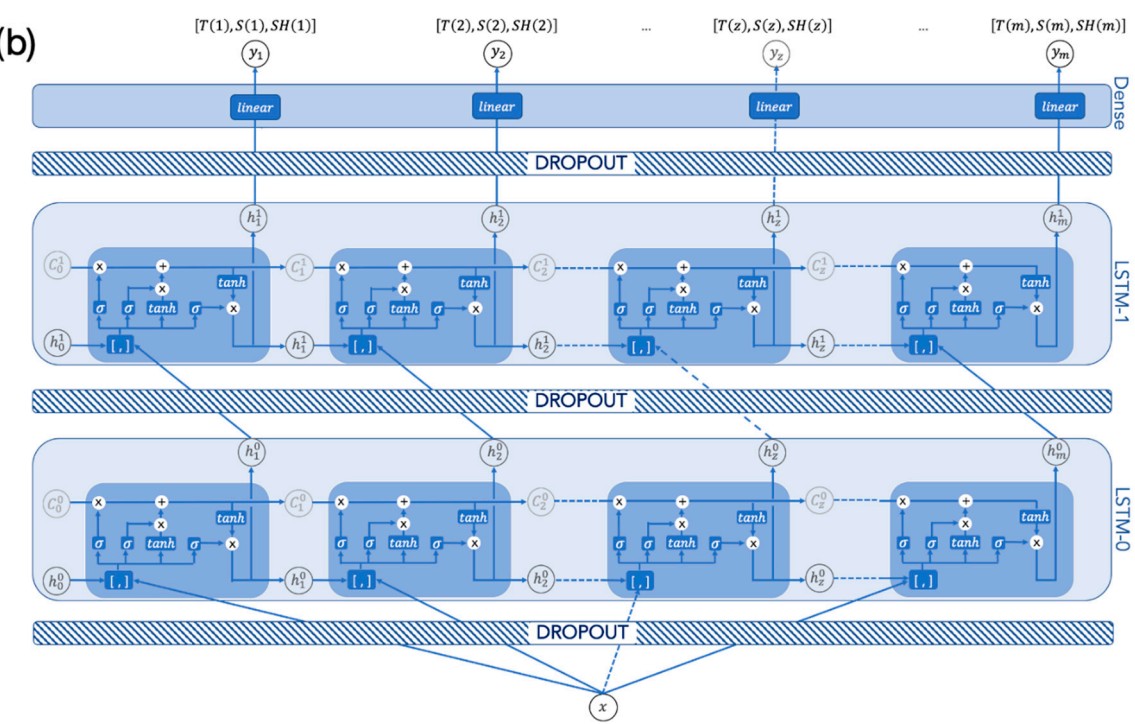

**Figure 2.** Diagram showing the elements of a single LSTM cell (**a**). Stacked LSTM model for the reconstruction of vertical hydrographic profiles (**b**).

### 2.7. Monte-Carlo Dropout

Standard dropout consists in randomly excluding a percentage of units during network training. Dropout provides a very efficient regularization strategy if applied during training, significantly reducing the risk of co-adaptation, thus limiting overfitting and improving model performance [55,56]. Moreover, dropout also provides an extremely simple and powerful approach to quantify a neural network uncertainty, if applied during both training and testing. In fact, running a regression neural network several times with dropout during testing generates different outputs for the same input. It was shown mathematically that these outputs are equivalent to Monte-Carlo sampling [42]. Hence,

ensemble mean and variance provide the network's output values and related uncertainty, respectively. During learning, 20% of the units were dropped here.

## 2.8. Code Availability

The LSTM/FFNN codes used here are released under the terms of the GNU General Public Licence v3 and available at the following address: https://github.com/bbuong/3Drec.

## 3. Results

The assessment of the techniques was performed by estimating temperature and salinity root mean squared differences (RMSD) with respect to randomly selected independent test data (Figure 3). To this aim, 15% of the 35,344 in situ profiles collected in the area (totalling 5125 profiles) were excluded from the network training. The data used for the technique assessment can be found at [57].

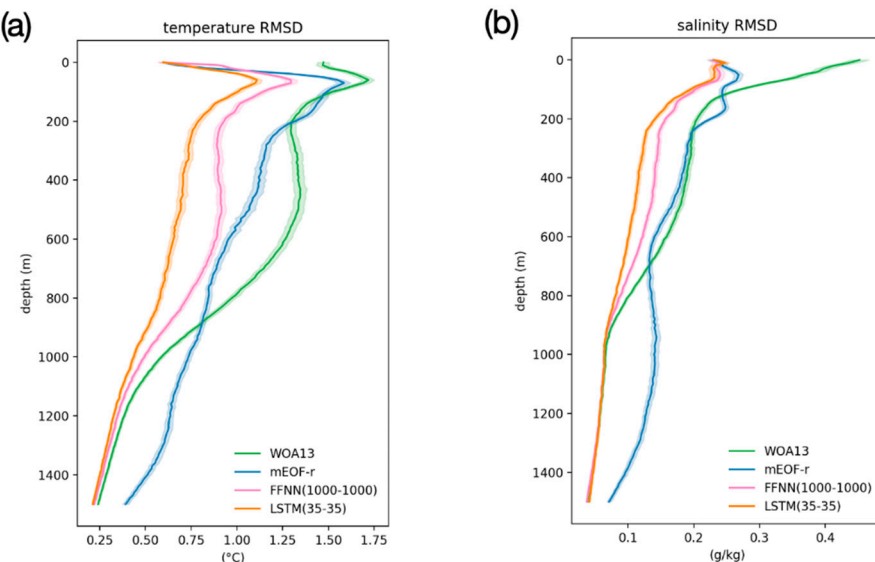

**Figure 3.** RMSD between temperature (**a**) and salinity (**b**) climatological and reconstructed profiles, estimated from independent test data. RMSD confidence intervals (one $\sigma$, displayed here as shadowed areas) have been estimated through a Monte Carlo approach—i.e., as the standard deviation of the statistics computed from 1000 resampling with replacement [58].

All techniques considered process anomalies with respect to WOA13 data, and are thus intended as corrections to the climatological profiles. Climatological temperature RMSD attains around 1.5 °C at the surface, reaches a maximum of up to 1.7 °C at ~100 m depth (at the base of the upper mixed layer) and then gradually decreases to the minimum error <0.25 °C at 1500 m, with a wide secondary maximum positioned around 500 m, characterized by errors >1 °C down to 800 m. The temperature retrieved with mEOF-r shows a moderate improvement in the upper 100 m and then in the 200–900 m layer, but then significantly degrades the climatology below 900 m. Conversely, both the deep FFNN (1000–1000) and the stacked LSTM (35–35) significantly improve the reconstruction all along the water column. Noticeably, LSTM (35–35) clearly outperforms any of the other methodologies, with a RMSD never exceeding 1 °C, and attaining below 0.75 °C already at a 200 m depth. Salinity RMSD show similar behaviours, with the climatological estimates reaching up to ~0.5 g/kg, and remaining above 0.2 in the upper 600 m, the mEOF-r displaying only a partial improvement in the upper 100 m (keeping its error close to that associated with the surface input data—i.e., around 0.25 g/kg), and FFNN (1000–1000) and LSTM (35–35) reducing the RMSD to almost one half, the LSTM (35–35) further improving in the 200–800 m layer (Figure 3b).

To further assess the model performance, we map the temperature and salinity differences (in absolute values) with respect to test observations, considering both mEOF-r and LSTM (35–35) reconstructions and focusing on two depths: 100 m and 500 m, which are the most impacted by upper mixed layer and thermocline depth variability, respectively.

At 100 m, both temperature and salinity retrieved with mEOF-r present large absolute differences in a wide area along the Gulf Stream, but also all along the southeastern border of the domain (Figure 4a,b). LSTM (35–35) displays quite smaller overall differences, which are also mostly confined to the areas where mesoscale variability is strongest (i.e., the Gulf Stream) and surface temperature and salinity input data are affected by the largest interpolation errors (Figure 4c,d). Indeed, the surface data used here are level 4 data—i.e., they are interpolated in space and time to fill in data voids present in satellite data due to clouds, rain, and/or other instrumental/orbital/coverage limitations. As such, they are less accurate in areas characterized by intense variability, as shown by the error fields provided within the products taken here as input (see Section 2.1). Errors in surface satellite data projection at depth thus clearly reflect the discrepancy between co-located satellite and in situ surface data in these highly energetic areas.

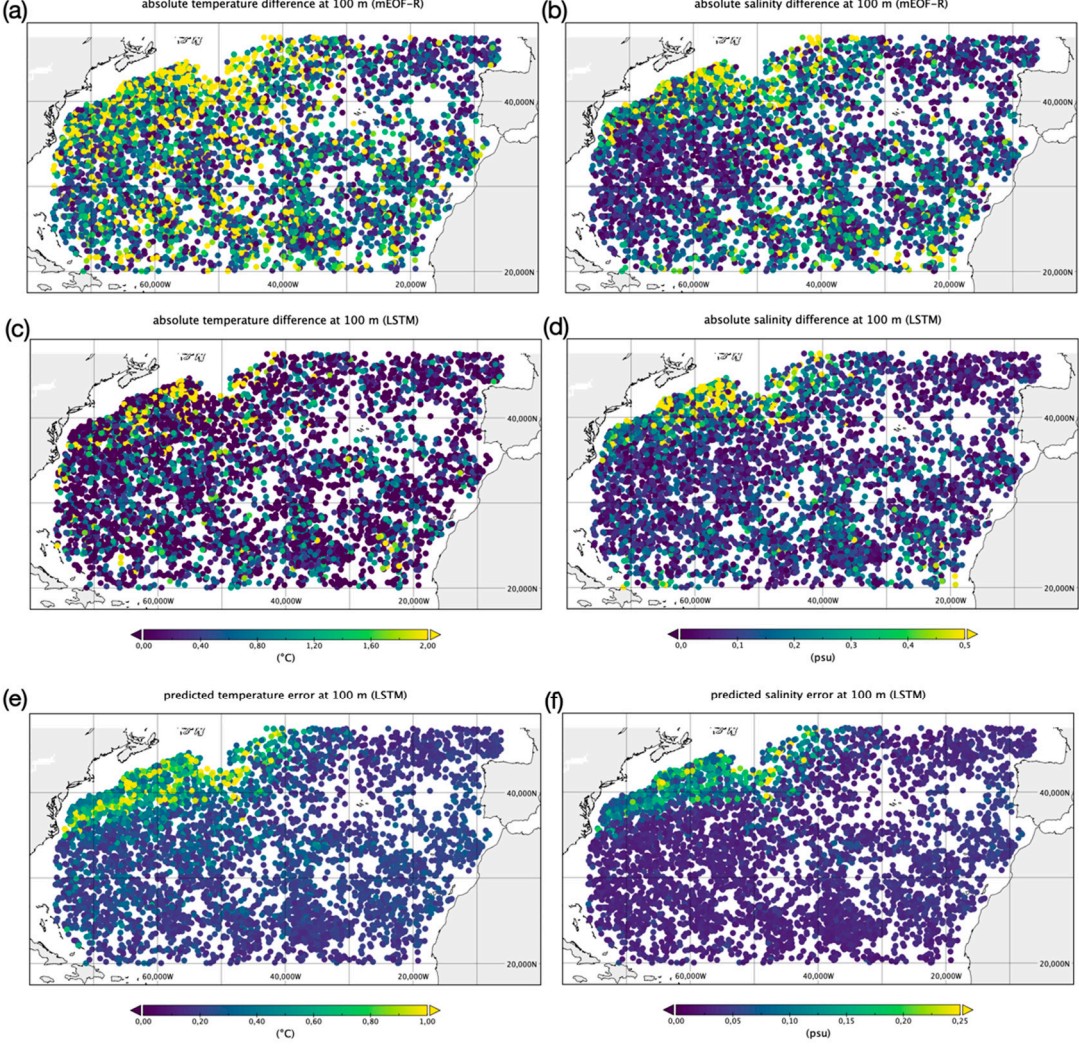

**Figure 4.** Absolute value of the differences between mEOF-r temperature (**a**) and salinity (**b**) and observed test data at 100 m depth, and corresponding LSTM (35–35) differences (**c,d**). Temperature (**e**) and salinity (**f**) predicted LSTM (35–35) reconstruction error at 100 m.

As anticipated, the neural network methods coupled with Monte-Carlo dropout present another significant advantage with respect to mEOF-r and similar statistical reconstruction techniques, being able to deliver not only retrieved values, but also associated uncertainties. Figure 4e,f thus present the RMSD predicted by the LSTM (35–35), which shows patterns that are consistent with the observed differences between test and synthetic reconstructions at 100 m. As the network has been directly trained with satellite-derived surface input data, it is likely that these patterns derive from the increased errors in surface satellite data along the western boundary of the domain. It cannot be excluded, though, that increased uncertainties also reflect the presence of more complex vertical patterns caused by dynamical factors, which could eventually lead to the need for additional predictor variables not considered (or not available) here.

At 500 m, the largest mEOF-r differences are mainly found over a wide area centred over the Gulf Stream, and also close to the North Africa eastern boundary upwelling system, though less evident than at 100 m (Figure 5a,b). LSTM (35–35) shows significantly smaller differences with respect to mEOF-r, which are mainly seen close to the core of the Gulf Stream (Figure 5c,d). Again, these patterns closely match with the predicted errors (Figure 5e,f).

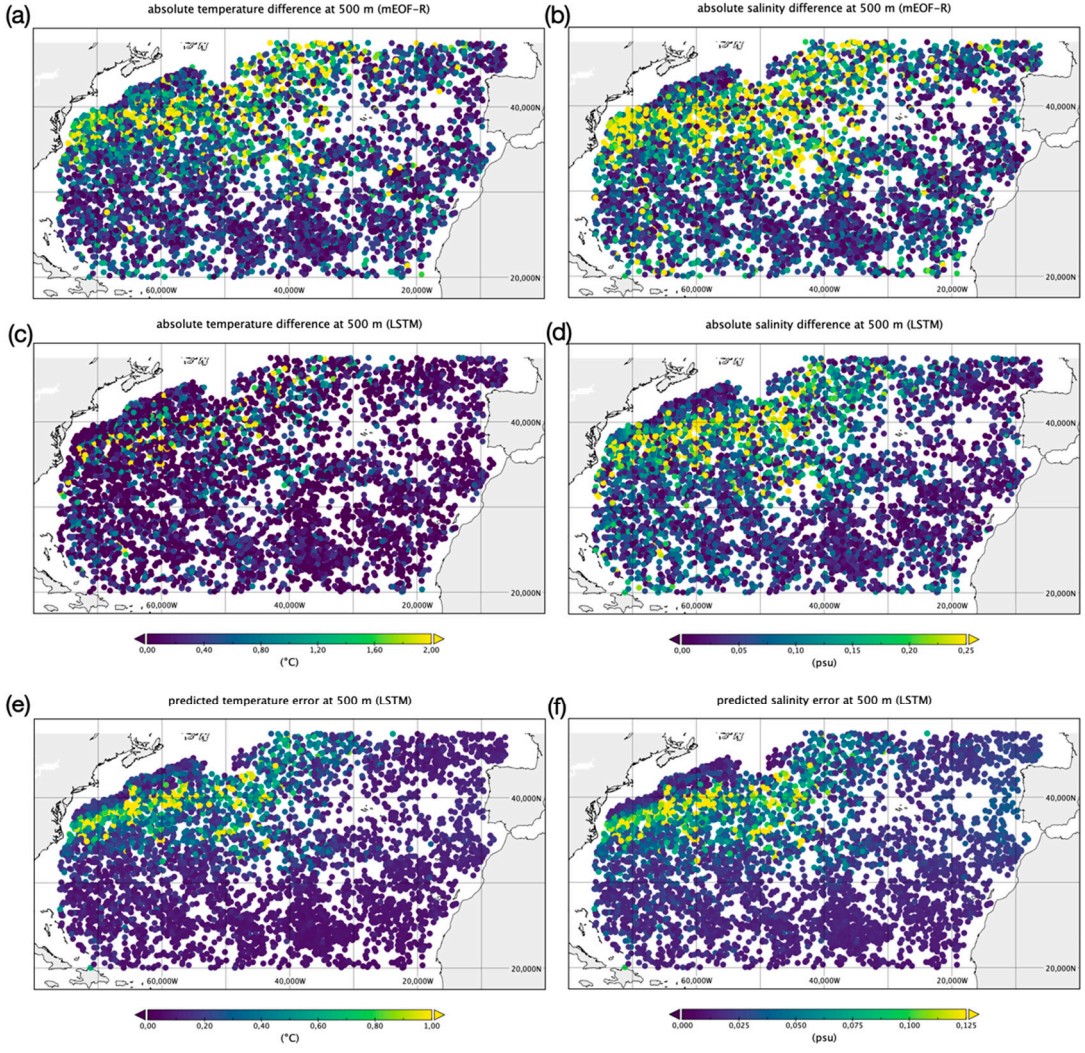

**Figure 5.** Absolute value of the differences between mEOF-r temperature (**a**) and salinity (**b**) and observed test data at 500 m depth, and corresponding LSTM (35–35) differences (**c**,**d**). Temperature (**e**) and salinity (**f**) predicted LSTM (35–35) reconstruction error at 500 m.

## 4. Discussion

Being able to monitor the ocean's interior structure is crucial to assess the impact of ocean dynamics on the Earth climate and marine ecosystems. However, available observations, acquired either from satellite sensors looking at the sea surface from space or from sparse in situ measurements of the water column, can only provide partial views of the 3D ocean state if analysed separately, due to their instrumental and sampling limitations. Several different techniques have thus been proposed to exploit satellite remote sensing data in combination with information extracted from in situ observations, either based on simple empirical correlations or more advanced statistical regressions [11–21]. Most of these methodologies are based on (multi)linear regressions or decompositions in empirical modes, which are not suited to properly disclose and model non-linear relations among available predictor and target variables. Conversely, artificial neural network modelling actually provides a wealth of almost unexplored algorithms that can be adapted to efficiently handle these kinds of problems. The deep learning algorithm presented here combines two advanced features that allow us to significantly improve the reconstructions with respect to those obtained from simpler empirical and feed-forward neural network architectures. First of all, it is based on a recurrent network scheme (Long Short-Term Memory) that is specifically designed to efficiently extract information from ordered sequences of data. While this is usually interpreted as a time series analysis tool, it can equally be applied to spatial sequences of data (such as vertical profiles here). Additionally, the algorithm is designed in such a way that a pre-defined percentage (20%) of neural connections is randomly excluded during both training and testing [50,51]. This dropout approach is equivalent to a Monte-Carlo sampling [38] and makes it possible not only to accurately retrieve target variables, but also to associate uncertainties with individual estimates.

The best performing LSTM model developed here includes two hidden (stacked) layers with 35 hidden units each. The RMSD of the profiles reconstructed with this technique vs. independent observations is reduced to as much as 50% with respect to reference reconstructions (climatological and mEOF-r profiles) and >20% with respect to a properly tuned deep feed-forward network (including 2 hidden layers with 1000 units each). Future work will thus focus on a more systematic application of LSTM to develop advanced 3D observation-based products, as well as on investigating the impact of additional predictor variables that could be estimated from available remote sensing data.

## 5. Conclusions

We have developed an innovative deep learning algorithm to project sea surface satellite observations at depth after learning from sparse co-located in situ hydrographic data. The proposed technique, based on a stacked Long Short-Term Memory neural network, coupled to a Monte-Carlo dropout approach, provides vertical profiles and associated uncertainties, outperforming both neural network reconstructions based on simpler feed-forward networks and multivariate EOF reconstruction. This technique will find immediate application for the development of a 3D product covering the North Atlantic in the framework of the European Space Agency World Ocean Circulation project (ESA-WOC). The work described here, however, covers only the development and assessment of the LSTM reconstruction methodology based on presently available data, as a new training of the network will be needed once updated ADT estimates will be made available by the project.

Remarkably, the adaptation of this technique to other areas/periods is easy and straightforward. Simultaneous availability of uncertainties associated with individual profiles also suggests that this deep learning methodology could be tested to extend present data assimilation approaches in numerical models by ingesting consistent remotely sensed sea surface data and synthetic profile estimates.

**Funding:** This research was partly funded by the European Space Agency through the World Ocean Current project (ESA Contract No. 4000130730/20/I-NB /WOC_Subcontract_Odl_CNR).

**Acknowledgments:** I thank Daniele Ciani for providing upsized ADT data remapped over the study area, and Michela Sammartino and Francesca Elisa Leonelli for helpful discussions at the initial stage of the study.

**Conflicts of Interest:** The author declares no conflict of interest.

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
