# Peer review of "A Deep Learning Network to Retrieve Ocean Hydrographic Profiles from Combined Satellite and In Situ Measurements"

_remotesensing, doi:10.3390/rs12193151_

Round 1

Reviewer 1 Report

Review of the manuscript remotesensing-939819

The manuscript is innovative and interesting with possibility to be highly cited paper.

I suggest the author to add the new section showing all models setup used in the manuscript, with parameters shown in details. The readers would like to make similar experiment.

Section 2.4: Please add the figure showing three EOFs. In situ observation s have different depths, resulting different size of state vector, please comment in the text. L157, please add some text showing configuration from [16].

Section 2.5: L197, please add some paragraph explaining “as in [31]”. L201 Please explain min-max algorithm.

In The introduction section, paragraph about feed-forward neural networks is missing. Also, (L68-74) I suggest to add one more paragraph about application of the Long Short-Term Memory network (not necessary in ocean science) .

Figure 3. Please explain how bootstrapping  was applied?

Author Response

Reviewer #1

The manuscript is innovative and interesting with possibility to be highly cited paper.

I thank the reviewer for their positive comment.

I suggest the author to add the new section showing all models setup used in the manuscript, with parameters shown in details. The readers would like to make similar experiment.

Few missing details have been added in the revised text (as per specific requests by the reviewer reported below), so that models’ architecture and main (hyper)parameters are fully described in sections 2.4-2.7. However, to further address the reviewer’s suggestion, and also to respond to a specific request by reviewer #3, I have now directly made the model code available on github, and included the link within the paper itself (Code availability section added at the end). This guarantees full reproducibility without making the reading too heavy with long lists of parameters.

Section 2.4: Please add the figure showing three EOFs.

As explained in this sub-section, there is no such thing as a unique set of EOF patterns to show, as mEOF-r computes different mEOFs for each of the overlapping sub-domains. Anyway, to respond to reviewer’s curiosity, L153 now includes reference to an example of mEOF modes in the North Atlantic from one of my previous papers (figure 3 in Buongiorno Nardelli et al., OS 2012).

In situ observations have different depths, resulting different size of state vector, please comment in the text.

As stated in L133 “The data considered here […] were interpolated through a spline on a regularly spaced vertical grid (with 10 m intervals).” So, that the state vector size is not changing at all (otherwise it would have been impossible to compute any EOF).

This has now been further clarified also at L151 “(normalized and interpolated on a regular vertical grid)”

L157, please add some text showing configuration from [16].

The paragraph has been slightly modified to clarify that the configuration is effectively described in the following lines (L163-L175).

Section 2.5: L197, please add some paragraph explaining “as in [31]”.

This is now explicitly defined in the text (L203-204).

L201 Please explain min-max algorithm.

A brief clarification on min-max meaning is now provided at L208-209 (see also https://en.wikipedia.org/wiki/Feature_scaling).

In The introduction section, paragraph about feed-forward neural networks is missing. Also, (L68-74) I suggest to add one more paragraph about application of the Long Short-Term Memory network (not necessary in ocean science) .

FFNN and LSTM theoretical aspects and implementation details are fully described in section 2. I do not agree on the need to duplicate that by adding more details in the introduction, where the overall problem and the general approach and objectives of the work are already presented in a balanced way. Not changed.

Figure 3. Please explain how bootstrapping was applied?

For all calculations, the significance of the statistics was estimated through a Monte Carlo approach as the standard deviation of the statistics carried out from 1000 resamples with replacement (Efron and Tibshirani, 1993). This is now explicitly written/referenced in the figure caption.

Reviewer 2 Report

A Deep Learning Network to Retrieve Ocean Hydrographic Profiles from Combined Satellite and In Situ Measurements by B.B. Nardelli

One of limitations to use remote sensing is to observe subsurface phenomena. The method demonstrated in this paper can be one of advanced algorithms for subsurface temperature and salinity reconstruction. Thus, I would like to recommend this study to be published in Remote Sensing after improving some of issues below.

Major

  • In Figure 4, explain why absolute values along western boundary are higher than those in other regions. Furthermore, there are some high differences (yellow and greenish spots) almost in everywhere (Figures 4a-d). Explain what makes those differences.  

Minor

  • Define FFNN(1000-1000), LSTM(35-35) of Figure 3 in the caption or manuscript. .

  • There are other important references, which might be worth to be cited in this study.

Su, H.;Wu, X.; Yan, X.-H.; Kidwell, A. Estimation of subsurface temperature anomaly in the Indian Ocean during recent global surface warming hiatus from satellite measurements: A support vector machine approach.

Su, H.; Yang, X.; Lu,W.; Yan, X.-H. Estimating Subsurface Thermohaline Structure of the Global Ocean Using

Surface Remote Sensing Observations. Remote Sens. 2019, 11, 1598.

Jeong, Y.; Hwang, J.; Park, J.; Jang, C.J.; Jo, Y.-H. Reconstructed 3-D Ocean Temperature Derived from Remotely Sensed Sea Surface Measurements for Mixed Layer Depth Analysis. Remote Sens. 201911, 3018.

Author Response

Reviewer #2

One of limitations to use remote sensing is to observe subsurface phenomena. The method demonstrated in this paper can be one of advanced algorithms for subsurface temperature and salinity reconstruction. Thus, I would like to recommend this study to be published in Remote Sensing after improving some of issues below.

 I do thank the reviewer for his positive and constructive comments.

Major 

  • In Figure 4, explain why absolute values along western boundary are higher than those in other regions. Furthermore, there are some high differences (yellow and greenish spots) almost in everywhere (Figures 4a-d). Explain what makes those differences.  

Thanks for giving me the opportunity to better clarify this. The LSTM/FFNN and mEOF-r reconstructions are based on the projection of satellite-derived sea surface values. The surface data used here are level 4 data, i.e. they are interpolated in space and time to get rid of data voids present in satellite data due to clouds, rain, and/or other instrumental/orbital/coverage limitations. As such, they are less accurate in areas characterized by intense variability, such as the Gulf Stream, as visible also in the error fields included in the products, in related product user manuals and dedicated publications by the data producers. Errors in surface satellite data projection at depth thus clearly reflect the discrepancy between collocated satellite and in situ surface data. This has now been better clarified in the manuscript (L322-338).

Minor 

  • Define FFNN(1000-1000), LSTM(35-35) of Figure 3 in the caption or manuscript. .

 This is now defined in the text and consistently used throughout the paper.

  • There are other important references, which might be worth to be cited in this study.

I thank the reviewer for suggesting these interesting papers as additional references. They are now cited in the manuscript.

Su, H.;Wu, X.; Yan, X.-H.; Kidwell, A. Estimation of subsurface temperature anomaly in the Indian Ocean during recent global surface warming hiatus from satellite measurements: A support vector machine approach.

Su, H.; Yang, X.; Lu,W.; Yan, X.-H. Estimating Subsurface Thermohaline Structure of the Global Ocean Using Surface Remote Sensing Observations. Remote Sens. 2019, 11, 1598.

Jeong, Y.; Hwang, J.; Park, J.; Jang, C.J.; Jo, Y.-H. Reconstructed 3-D Ocean Temperature Derived from Remotely Sensed Sea Surface Measurements for Mixed Layer Depth Analysis. Remote Sens. 2019, 11, 3018

Reviewer 3 Report

I am not familiar with neural network and AI but the method described in this manuscript seems to work well.  I hope the author provides the detailed technique in an open source website so that the community can make use of this method.  The author may refer to the following references since they are different from the current method but aimed at the same objective:

Takano et al. (2009) J. Atmos. Oceanic Tech, 26(12) 2655-

Uchiyama et al. (2017) Atmosphere-Ocean, DOI:10.1080/07055900.2017.1399858

Author Response

I am not familiar with neural network and AI but the method described in this manuscript seems to work well.  I hope the author provides the detailed technique in an open source website so that the community can make use of this method. 

Following the reviewer’s suggestion, I have now made the model code available on github, and included the link within the paper itself (Code availability section added at the end).

The author may refer to the following references since they are different from the current method but aimed at the same objective:

Takano et al. (2009) J. Atmos. Oceanic Tech, 26(12) 2655-

Uchiyama et al. (2017) Atmosphere-Ocean, DOI:10.1080/07055900.2017.1399858

I thank the reviewer for suggesting these interesting papers as additional references. They are now cited in the manuscript.

Round 2
